# Very-Long-Chain Acyl-CoA Dehydrogenase Deficiency: Family Impact and Perspectives

**DOI:** 10.3390/ijns9040053

**Published:** 2023-10-06

**Authors:** Sarah Crawford, Elizabeth Sablon, Nadia Ali, Ami R. Rosen, Patricia L. Hall, Juanita Neira Fresneda

**Affiliations:** 1Cincinnati Children’s Hospital Medical Center, Cincinnati, OH 45229, USA; 2Department of Human Genetics, Emory University, Atlanta, GA 30322, USA; 3Mayo Clinic, Rochester, MN 55905, USA

**Keywords:** VLCADD, caregiver, barriers, medical diet, rhabdomyolysis, cardiomyopathy, metabolism

## Abstract

Very-Long-Chain Acyl-CoA Dehydrogenase Deficiency (VLCADD) is a fatty acid oxidation disorder characterized by the decreased ability of the enzyme very-long-chain acyl-CoA dehydrogenase to break down fatty acids with 14 to 20-long carbon chains. The resulting clinical manifestations are variable in severity and include hypoketotic hypoglycemia, rhabdomyolysis, and cardiomyopathy. Treatment can consist of limiting the dietary intake of long-chain fatty acids, the prevention of fasting, and the supplementation of medium-chain fats. This study, conducted in the context of a 5-year long-term follow-up on VLCADD, evaluates how the diagnosis of this fatty acid disorder impacts the family, specifically as it relates to the medical diet and barriers to care. Caregivers (*n* = 10) of individuals with VLCADD responded to a survey about how VLCADD potentially impacts their family. The review included the clinical outcomes of the patients (*n* = 11), covering instances of rhabdomyolysis, cardiomyopathy, and hospitalizations related to VLCADD. Families affected by VLCADD experience barriers to care, including difficulties with finances, ability to work, and access to nutrition.

## 1. Introduction

Very-long-chain acyl-CoA dehydrogenase deficiency (VLCADD) is a recessive inborn error of fatty acid oxidation caused by an enzymatic deficiency in the first step of beta oxidation in the mitochondria [1]. The enzyme is encoded by *ACADVL*. Very-long-chain acyl-CoA dehydrogenase is responsible for the breakdown of fatty acids with 14 to 20-long carbon chains. The enzyme deficiency mainly impacts tissues requiring high amounts of energy, including the liver and muscle. During periods of increased metabolic stress, such as illness or exercise, hypoketotic hypoglycemia and rhabdomyolysis can occur when fat stores become the preferred energy source [2]. Long-chain fatty acids can also deposit in the heart, contributing to cardiomyopathy in the long term. To prevent metabolic decompensations, patients are advised to restrict dietary long-chain fatty acid intake and avoid fasting [3].

The Southeast Regional Genetics Network (SERN) and Genetic Metabolic Dieticians International (GMDI) published a comprehensive protocol for treating patients with VLCADD based on the product of a 2020 workgroup consisting of experts in the field and the evaluation of research articles. It recommends that all individuals, regardless of their clinical presentation, should avoid fasting. However, the protocol emphasizes the variability in this condition, especially due to the inclusion of this condition in newborn screening; because of this, the percentage of fat in the diet should differ based on age, weight, and symptoms. The recommended limits of dietary long-chain fat intake can range from 10–40% of total intake. Many patients use supplements containing medium chain triglycerides to bypass the need for very-long-chain acyl-CoA dehydrogenase [4].

Newborn screening has altered the outcomes of this condition in a positive way. It has allowed the identification of children earlier in life, as well as the initiation of treatment earlier on to prevent complications. VLCADD was added to the Recommended Uniform Screening Panel in the United States in 2006 [5]. With the advent of newborn screening, there are three times as many cases being detected, including more individuals with mild VLCADD [6]. The prevalence is now estimated to be approximately 1 in 31,500 births [7], whereas previously the prevalence was estimated to be approximately 1 in 110,000 births [6]. Studies have evaluated the outcomes of patients with VLCADD with a spectrum of severity regarding its clinical manifestations [3,8,9]. This study adds to that work.

While previous studies have provided clinical details on VLCADD, there is limited literature on the impact of VLCADD on the family. From other studies on families that have children with metabolic conditions, there is evidence that a medical diet negatively impacts caregivers’ quality of life. An example can be found in Tyrosinemia Type I, as caregivers identified that the dietary treatment for the condition caused an emotional impact [10].

The present study was conducted as part of a 5-year Georgia long-term follow-up project (GA-LTFU) that is currently ongoing and is assessing the health outcomes of children regarding three genetic disorders: VLCADD, MCADD, and SCID. Part of the GA-LTFU includes a parent database that documents barriers to care. The database collects detailed clinical information on cases of VLCADD, including variants, clinical outcome measures, diet details, and diet adherence. The present sub-study sought to identify factors outside of the condition itself, including time availability, insurance, financial barriers, and parents’ perception of the severity of illness, that can have an impact on families affected by VLCADD. By identifying barriers to care, efforts can be made to improve the experience and care for these families.

## 2. Materials and Methods

This study was approved by the Emory University IRB (IRB ID: IRB00106320). The study team developed a questionnaire using modified questions from the Pediatric Inventory for Parents (PIP) and a caregiver survey on Tyrosinemia [10,11]. This questionnaire was made in response to preliminary data from the parent study suggesting barriers to care in this population. The questionnaire was targeted towards caregivers and included demographic questions, questions about stress levels, how VLCADD has or has not impacted the family, and potential barriers to care for VLCADD (see Appendix A).

The inclusion criteria comprised having a child with a confirmed diagnosis of VLCADD. Families either must have received care in the past or were currently receiving care at the time of recruitment in Georgia for the treatment and monitoring of VLCADD. To control for broad age ranges, only caregivers of children born between 2014 and 2021 were eligible. The Georgia Newborn Screening database was filtered from 2014–2021, and the Emory Metabolic Genetics Clinic also provided a list of current patients attending clinic for VLCADD that had Newborn Screening in other states. This revealed 30 eligible participants in total.

The questionnaire was open from February 2022 to August 2022. The questionnaire was conducted via phone and during patient appointments. Ten caregivers provided verbal consent to participate in the study for a total of eleven patients with VLCADD (*n* = 11). Two children (patient #6 and patient #7) were siblings. The response rate was 36.6% (11/30). In total, 40% (4/10) of surveys were completed in the presence of a geneticist, as these 4 surveys were collected during appointments. All caregivers who participated in the survey were mothers. A grant from the Georgia Association for Genetic Counselors enabled compensation in the form of a USD 15 retail gift card.

To inform the survey responses, a retrospective chart review was completed on each patient through the electronic medical record. The data we collected included genetic test results, history of creatine kinase (CK) elevation, diet compliance based on the last clinic note, and clinical outcomes (presence or absence of cardiomyopathy, rhabdomyolysis, and hospitalizations/ER visits). All data collected were kept confidential with an encrypted system in the LTFU database.

Data analysis involved descriptive statistics for demographics. We conducted Fisher’s exact tests for associations between the clinical outcomes and caregiver responses. However, the rules of the test were violated due to the small number of participants in the cohort, so we did not include these results in this publication. We focused on trends as opposed to statistical associations.

## 3. Results

### 3.1. Participants

Ten caregivers participated in the study and completed the questionnaire. All were mothers of at least one child with VLCADD. Demographics are shown in Table 1.

### 3.2. Survey Responses

The responses to the caregiver survey are summarized in Table 2 and Figure 1, and have been divided according to topic. All ten caregivers indicated that insurance did cover their needs regarding their child’s VLCADD, with two instances in which insurance had not done so in the past. Additionally, ninety percent (9/10) indicated that they receive services from a Metabolic Genetics provider as often as recommended. One caregiver indicated two responses: they see a Metabolic Genetics provider only when their child gets sick and “when I remember”. A dietary plan in school was relevant for six patients in the cohort.

In total, 30% (3/10) of caregivers had more than one child with VLCADD (Table 1). These caregivers did not agree that VLCADD was difficult to manage. Additionally, these three caregivers indicated that VLCADD impacted their ability to access care in some way. For the seven caregivers that had one child with VLCADD, 57% (4/7) agreed that it was difficult to manage (Figure 1).

In total, 50% (5/10) of caregivers had other children who were not affected by VLCADD (Table 1). One caregiver out of these five agreed that VLCADD was difficult to manage. All five indicated in at least one way that VLCADD impacts access to care.

Out of the eight caregivers who indicated an impact on access to care, 62.5% (5/8) indicated multiple impacts. For the three caregivers who chose “causing financial constraints”, one also indicated that VLCADD impacted their ability to work (Figure 1).

### 3.3. Clinical Status from Chart Review

The following symptoms and variants were obtained from the chart review via clinic notes, CK values, and genetic testing reports. Of the seven caregivers who had increased emotional stress (Figure 1) caused by VLCADD (for a total of eight children with VLCADD), 37.5% (3/8) of children had experienced a documented instance of rhabdomyolysis and 75% (6/8) had a documented ER visit or hospitalization related to VLCADD. Of the three caregivers who did not indicate emotional stress (for a total of three children with VLCADD), 66.6% (2/3) of children had both a documented instance of an ER visit or hospitalization related to VLCADD and rhabdomyolysis, and one child had neither a documented instance of an ER visit or hospitalization nor rhabdomyolysis.

Figure 2 shows clinical outcomes of individuals with VLCADD. Adherence to the medical diet was determined through clinical notes. There were two patients (siblings) who were not adhering to the medical diet at the time of the chart review. Their caregiver strongly disagreed that VLCADD was difficult to manage, but indicated increased emotional stress from the condition for both children. For the nine patients who were adhering to the diet at the time of the chart review, 66.6% (6/9) of these caregivers reported increased emotional stress.

Instances of rhabdomyolysis were based on both clinical notes and lab data. Individuals with at least one instance of a hospitalization/ER visit related to VLCADD were included in the “yes” category. The presence or absence of cardiomyopathy was determined via echocardiogram. Two patients had not had an echocardiogram at the time of the chart review, and were presumed to not have cardiomyopathy and were put in the “no” category for cardiomyopathy, as shown in Figure 2. However, the lack of an echocardiogram does not confirm that cardiomyopathy is truly absent. Those receiving ongoing care from a metabolic genetics provider were put in the “yes” category for attending clinic.

Responses to the open-ended question at the end of the survey resulted in various topics, as depicted in Table 3. The responses received relate to the impact of VLCADD. Seven caregivers provided a response to the open-ended question.

## 4. Discussion

This study adds to the limited research on the quality of life for caregivers of children with long-chain fatty acid oxidation disorders, offering a specific perspective on having a child with VLCADD. The data identified the barriers to care that families who are affected by VLCADD experience, including difficulties with finances, ability to work, and access to nutrition. While the limited number of subjects in our cohort prevents us from making broad generalizations for individuals with VLCADD and their caregivers, pilot conclusions can be made on the basis of the reported experiences at a single center providing care for VLCADD in Georgia.

Eighty percent (8/10) of caregivers indicated that VLCADD has impacted their family’s ability to access care in some way, suggesting that factors outside of the condition itself have an influence on clinical care. While the questionnaire did not specify whether caregivers were currently experiencing an impact, some verbally expressed during the administration of the questionnaire that they no longer experienced this impact, but did when their child was first diagnosed or for a period of time. This suggests that barriers to care can be transient, but also enables the conclusion that access to care could have similar interruptions. Further, visits to clinics with metabolic health care providers, including dieticians, can help families with adjusting to diagnosis [12]. A recent study evaluated the quality of life for adult patients and their caregivers for multiple long-chain fatty acid oxidation disorders. In their cohort, 70% of caregivers experienced impacts on their work due to their child’s LC-FAOD. Overall, this study found lower quality of life measures compared to the average American in both the physical and mental health domains [13].

Even though 7/10 caregivers indicated that VLCADD increased emotional stress for the family, there was not a trend showing worse clinical outcomes for those eight affected individuals versus the three individuals who had caregivers that did not indicate emotional stress. Out of those seven who indicated increased emotional stress, three out of those seven did not agree that their child or children’s VLCADD was difficult to manage. This is perhaps explained by an ability to adjust to the diagnosis despite feeling stress.

Some caregivers had multiple children with VLCADD. As none of these caregivers agreed that VLCADD was difficult to manage, this suggests that caregivers can adjust to the diagnosis, including the demands of the medical diet and recognizing symptoms. This is consistent with the open-answer response from a mother who said, “Before I used to have a lot of stress and felt a lot of pressure, but now with my second child I feel relieved and have learned about it” (Table 3).

The presence of increased stress within the family is consistent with previous research on inherited metabolic disorders. A qualitative study on coping with inherited metabolic disorders conducted in Canada identified stressors directly related to adjusting to the medical diet. Caregivers identified stress related to adjusting to the “new normal”, coordination with the school regarding their child’s diet, and the inevitability of their child feeling left out of certain social situations [12]. Another qualitative study focusing solely on long-chain fatty acid metabolic conditions identified similar stressors. The themes revealed through the interviews in this study were related to social domains, such as being the recipient of bullying in school due to a special diet, and related to lifestyle modifications, such as the need to avoid activities or plan ahead because of the risk of metabolic crisis [14]. Many of these stressors were identified in the open-answer responses given in the present study.

The open-answer responses in the present study provide a glimpse into the potential social impacts faced by children affected by VLCADD. It also shows how parents are responsible for their child’s diet at school and for educating peers, family, and friends. In future studies on VLCADD, qualitative research focusing on only VLCADD would be helpful regarding the identification of specific barriers and impacts on children and their families, such as the experiences related to rhabdomyolysis and ER visits.

The small cohort was a limitation in this study, a common finding when studying rare diseases. This resulted in a descriptive presentation of data, rather than a focus on statistical significance. With any targeted survey, a source of bias is only considering answers from those who complete the survey. This may not be representative of the population of families affected by VLCADD. For example, families who were either significantly impacted or, conversely, not as impacted by VLCADD, may have chosen not to answer the survey. For those who did complete the survey, they may have chosen answers related to their child’s medical condition that are incorrect. This can be caused by recall bias or the availability of an answer choice that looks like the “best” answer choice. These data are also limited by the recruitment methods used, which resulted in exclusively mothers as caregivers completing the survey.

This study acknowledges that it is possible for families to experience hardships and not report increased stress. The answer choices for some of the questions are subjective. For example, for the question about how the condition impacts the family, many people taking the survey answered the question by saying that there was stress when the child was first diagnosed. Four surveys were completed in the presence of a geneticist and/or dietician. This may have caused caregivers to answer differently, especially to questions relating to compliance. Because the survey was short and had closed-ended questions, expanding on answers was performed in the “open-ended” section. Some responses may have been additionally influenced, as the survey asked about stress and this study took place during the COVID-19 pandemic. The answers were a snapshot in time and stress can change based on the phase of life, condition, and context of the family.

## 5. Conclusions

There are impacts on the family outside of VLCADD itself, including social factors and barriers to care. The potential strategies used to address barriers to care can include enhanced support and awareness from entities that support the care of affected children, such as medical centers, government assistance programs, and school administrations. The importance of receiving ongoing care from providers with an extensive knowledge of this condition and its treatment is also supported by this study. Acknowledging the heightened responsibility placed on parents and the affected child to adhere to the medical diet is a significant part of caring for this population. Additionally, the medical community should acknowledge the resiliency of these families despite the hardships they face. The expansion of this pilot data to other centers could further help this population with stressors outside of the condition itself.

## Figures and Tables

**Figure 1 IJNS-09-00053-f001:**
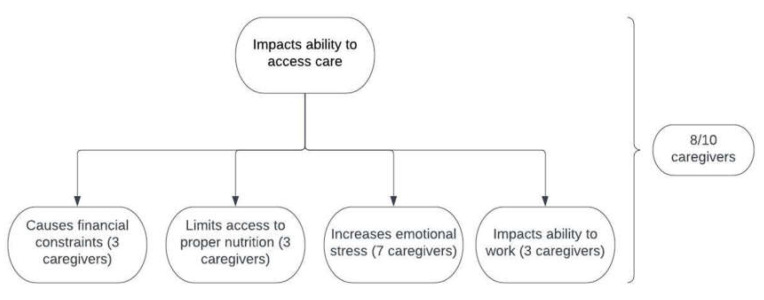
Responses from the caregiver survey regarding how VLCADD impacts access to care. Caregivers who answered that VLCADD impacts access to care chose one or more of the following impacts: causes financial constraints, limits access to proper nutrition, increases emotional stress, and/or impacts ability to work.

**Figure 2 IJNS-09-00053-f002:**
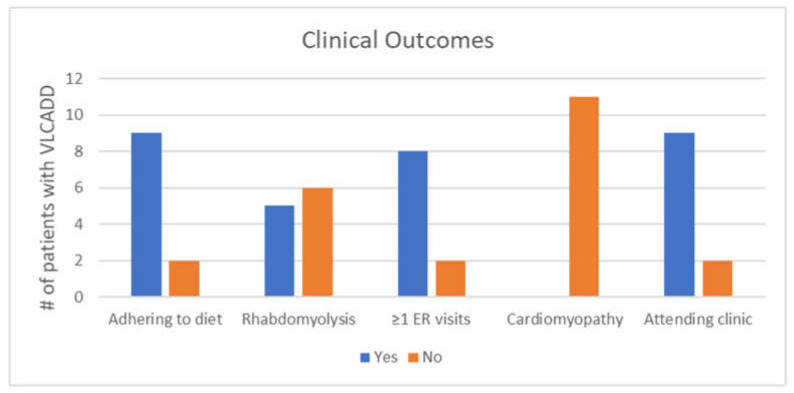
Clinical outcomes based on the chart review of 11 individuals with VLCADD.

**Table 1 IJNS-09-00053-t001:** Demographics of caregivers and participants with VLCADD.

Variable	Level	*n* = 10 (Caregivers), *n* = 11 (Patients)
Age range of caregivers	31–35 years	6
	36–40 years	3
	41–45 years	1
Biological Sex	Female	10
Number of children with VLCADD per household	1 child	7
	2 children	2
	3 children	1
Number of children per household	1 child	5
	2 children	1
	≥3 children	4
Age range of patients	0–2 years	3
	3–5 years	4
	6–8 years	4

If families had more than one child with VLCADD, these children were included in this study only if they were born between 2014–2021.

**Table 2 IJNS-09-00053-t002:** Responses to caregiver survey on impact of VLCADD on the family.

Survey Topic	Answer Choice	*n* = 10
VLCADD difficult to manage	Strongly agree/agree	4 (40%)
	Neutral	1 (10%)
	Strongly disagree/disagree	5 (50%)
Insurance coverage	Yes	10 (100%)
	No	0 (0%)
Frequency of seeing a metabolic genetics provider	As often as recommended	9 (90%)
	When I remember	1 (10%)
	Only when child gets sick	1 (10%)
Dietary plan at school	Yes	4 (66%)
	No	2 (33%)
Pediatrician aware of diagnosis	Yes	9 (90%)
	No	1 (10%)

**Table 3 IJNS-09-00053-t003:** Caregiver response to the open-ended question categorized according to topic.

	Response from Caregiver
Education	“I try to inform people as much as possible because it’s so rare.”
Initial Stress of Diagnosis	“Before I used to have a lot of stress and felt a lot of pressure, but now with my second child I feel relieved and have learned about it.” (Has as an older child with VLCADD) “In the beginning, it was very difficult. How do I differentiate between newborn fussiness and [symptoms of VLCADD]?”
Diet	“There is definitely a daily impact. Most families can stop and grab something to eat but we can’t do that...we have to prepare if we’re going to be out of the house.”
	“I send in a notecard of what he’s supposed to have for lunch each day.”
	“Instead of [school] providing her lunch, I provide her lunch.”
Healthcare Coverage	“Insurance used to not cover the care needed, but now it does.” Caregiver previously qualified for a government program involving childcare from a medically trained worker, but did not qualify after separation from her partner because she did not live at her child’s address.
COVID-19	COVID-19 prevented family from going to doctor’s appointments and are now “playing catch up” with child with VLCADD for appointments.

## Data Availability

Data from this study can be accessed by contacting the corresponding author.

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
