# Peer review of "Very-Long-Chain Acyl-CoA Dehydrogenase Deficiency: Family Impact and Perspectives"

_2409-515X, 2023, doi:10.3390/ijns9040053_

Round 1
Reviewer 1 Report
This is a well written description of the impact on caregivers of having a child with VLCADD. You acknowledge the small size of the cohort and other sources of potential bias which limit its generalisability. I have no concerns about the conduct of the study. However the themes arising from it are familiar to anyone working within IMD which means it lacks novel ideas or original findings. I would like to see you use your data to address more difficult questions around the impact of newborn screening.
Your study has the potential for novelty if you make more of the variant analysis data presented in the supplementary material. You even state that 'variant analysis was conducted for furthering understanding of the variant in relation to the patient's clinical presentation' (Lines 102-7) but make no further reference to it in the main text. I suggest considering re-writing as a 10 or 11 patient case series linking each patient's genetics to the clinical outcomes and caregiver responses. Examples of themes to consider in your report may include (but not be limited to):
1. How do answers from caregiver of patient 11 with two possible VUS differ from answers from eg patients 1 or 2 who have two pathogenic variants. For example is the emotional impact the same irrespective of predicted or actual severity of the condition?
2. Compare the emotional impact on caregivers of the two patients who do not adhere to diet with those who do adhere to diet. How does the severity as predicted by genetics and as evidenced by clinical outcomes relate to diet adherence?
3. Rhabdomyolysis: can you relate severity as predicted by gene variants to incidence of rhabdomyolysis? What were the other presentations to ER for - was it parental anxiety or a clinical VLCADD complication? How do responses of caregivers of children who have had rhabdomyolysis differ from those who have not?
Newborn screening is often hailed as a triumph of modern medicine but we do have to acknowledge it has downsides - notably identification of people with conditions who may never have become unwell.
Whilst you do not have sufficient data to make any broad generalisations, you could use your data as a pilot study to to a larger multi-centre study to help us as health professionals answer the big questions that will arise from newborn screening. In particular your data could be a prelude to a better understanding of the phenotypic variation of the condition to help us to provide more tailored advice depending on a patient's individual combination of pathogenic, likely pathogenic or VUS variants. We may also be able to use your data to improve our understanding of the relative impact on caregivers' wellbeing of having a child who needs to follow a medical diet (and consider whether it is even necessary for all VLCADD patients follow a medical diet) vs. that associated with having a child with a condition that could acutely decompensate. This would help us to prioritise interventions towards keeping patients well whilst not over-medicalising and generating undue anxiety.
Minor comments
Line 163, Figure 1. is not referenced in the text and also does not appear to show how VLCADD impacts 'access to care' but rather how it impacts on daily life including quality of life
Line 206 selection bias is acknowledged but it is just as likely that families who are more impacted may also be MORE likely to answer a survey
Reviewer 2 Report
On first glance, the title suggests that there is going to be a correlation of the impact information with some genotypic information for these families. This was clearly not borne out through the text, as the genotypic information was not even mentioned within the discussion. There is a significant body of literature describing severity estimates either from clinical information or from functional studies for various sequence variants. The inclusion of the genotypic information, as currently presented, does not add to this manuscript. Was there any correlation of impact on the family with potential severity of the sequence variants?
If genotypic information is going to be included in a subsequent draft of this manuscript, organize it, and Table S2, by variant first to avoid the unnecessary lines in the table where the same variants are present in sibs or common variants occur more than once within your sample. Organization by a patient ID adds nothing since there is no discussion of these variants in respect to any of the impact information or about their uniqueness within the current draft. This reviewer would strongly urge you to remove variant information as it is outside of the theme of the majority of the paper.
Did you have any information regarding whether or not each caregiver also was responsible for the care of a child without VLCAD deficiency? Were there differences in answers to the discrete or open-ended questions based on prior rearing experiences?
There are some missed opportunities for comparison of 2 subgroups, those with more than 1 child affected by this disease in contrast to those with a single child. Any all/none difference might be significant in larger populations than that of your pilot study.
Figure 1 is not individually cited within the text. The information in that figure could be included as part of table 2.
For the responses to Survey question 6: were any of the affirmative answers linked by caregiver, i.e. financial constraints and ability to work?
How did you let caregivers, whose family member’s appointments were outside of the February-August 2022 window, know about the study. This would help clarify the potential ‘n;’. Since you knew that some of the children were sibs, does affect the real ‘n’, as 30 eligible VLCAD children represented 29 different caregivers.
Can you explain why your IRB allowed verbal consent for a study involving detailed review of medical records? Since very rare data is currently included in the manuscript, such as unique variants, these individuals have a higher probability of being identified.
The information in lines 135-139 should be discussed in more detail in lines 178-184.
Be careful with the second paragraph of the discussion as you introduce new data which is not in the results section. The verbally expressed comments of caregivers are not otherwise documented. Did all caregivers make comments or only selected ones with different responses to some of the other questions?
You need to consider Kruger E et al 2022 Mol Gen Metab Reports in the discussion of other qualitative/semiquantitative impact studies on long chain fatty acid oxidation disorders. A comparison of adult patients, caregivers in your cohort would be appropriate.
In line 211: This reviewer would point out that it was exclusively mothers who completed the survey rather than a predominance of mothers.
The information in four surveys were completed in the presence of the geneticist at her dietitian needs to be in the methods section as this goes to reproducibility of the study design, or its flaws.
More complete citation is needed for Franklin by genoox (including the web address) as this may not be familiar to all readers. Similar concern exists for Varsome.
How often were echocardiograms obtained on these patients? Cardiomyopathy might be episodic and only associated with a decompensation. Presuming that the 2 patients, who did not have an echocardiogram, did not have cardiomyopathy needs an explanation based on genotype-phenotype information or family history for an older sibling.
Minor:
Line 54: Put citations numbers in ascending order.
Line 67: use ‘substudy’ or ‘sub-study’
Line 142: Right out laboratory rather than use ‘lab’.
Line 191: Does the ‘these’ refer to citation 14 citation 15 or both?
More consistency is needed in terms of the citations. Look at the formatting of 4 and 16 as examples. Doi should be included for references were possible, including labral 1
Supplementary materials Table S1, Item 11: 504 plans would be familiar only to some US readers. This needs to be explained in a footnote.
Round 2
Reviewer 1 Report
Thank you for addressing my comments. By removing the variant data you've shifted the emphasis of the study towards a qualitative assessment of the impact on caregivers of having a child or children with VLCADD. The study now has a clear focus and helpful message for health professionals whilst acknowledging the limitations (small sample size, one centre, one healthcare system) and cautioning against generalisability.
Author Response
Thank you very much for your helpful suggestions that improved this manuscript!
Reviewer 2 Report
This revised manuscript is much more focused which has allowed you to emphasize the need for larger studies of the impact of a diagnosis on the family. This manuscript is a well described pilot set of observations which I hope will prompt readers to collaborate in a much larger study of a more diverse cohort of families facing the same issues.
Author Response
Thank you very much for your helpful suggestions which improved the manuscript!